# Effects of Vitamin D Supplementation on Lipid Profile in Adults with the Metabolic Syndrome: A Systematic Review and Meta-Analysis of Randomized Controlled Trials

**DOI:** 10.3390/nu12113352

**Published:** 2020-10-30

**Authors:** Fatme AlAnouti, Myriam Abboud, Dimitrios Papandreou, Nadine Mahboub, Suzan Haidar, Rana Rizk

**Affiliations:** 1Department of Health Sciences, College of Natural and Health Sciences, Zayed University, Abu Dhabi 144534, UAE; myriam.abboud@zu.ac.ae (M.A.); dimitrios.papandreou@zu.ac.ae (D.P.); 2Department of Nutrition and Food Sciences, Faculty of Arts and Sciences, Lebanese International University, Beirut 14404, Lebanon; nadine.baltagi@liu.edu.lb (N.M.); suzan.haidar@liu.edu.lb (S.H.); 3Department of Health Promotion, Faculty of Health, Medicine and Life Sciences, Maastricht University, 6200 MD Maastricht, The Netherlands; 4Institut National de Santé Publique, d’Épidémiologie Clinique et de Toxicologie (INSPECT-Lb), Beirut 14404, Lebanon; rana.rizk@inspect-lb.org

**Keywords:** vitamin D supplementation, metabolic syndrome, dyslipidemia, cholesterol, triglycerides, adult, systematic review, meta-analysis

## Abstract

Background: Metabolic syndrome (MetS) increases the risk of cardiovascular disease, with atherogenic dyslipidemia being a major contributing factor. Methods: A systematic review was conducted following the Preferred Reporting Items for Systematic Reviews and Meta-Analyses (PRISMA) statement to assess whether vitamin D supplementation (VDS) alleviates dyslipidemia in adults with MetS. Scientific databases (PUBMED, MEDLINE, CINAHL, EMBASE, Cochrane Library, ClinicalTrials.gov, International Clinical Trials Registry Platform) and the gray literature were searched for randomized controlled trials of VDS, reporting on blood lipids. A narrative review, meta-analyses, sensitivity analyses, and appraisal of the risk of bias and overall quality of evidence produced were conducted. Results: Seven studies were included, and four were meta-analyzed. The risk of bias was generally low, and the final quality of evidence was low or very low. VDS, whether in high or low dose, significantly increased endline vitamin D blood levels; did not affect total, low-density, high-density cholesterol levels, and novel lipid-related biomarkers; yet, significantly increased triglycerides (TG) levels compared with placebo (MD: 30.67 (95%CI: 4.89–56.45) mg/dL; *p* = 0.02 for low-dose VDS; and MD: 27.33 (95%CI: 2.06–52.59) mg/dL; *p* = 0.03 for high-dose VDS). Pertaining heterogeneity was high (I^2^ = 86%; and I^2^ = 51%, respectively), and some included studies had significantly higher baseline TG in the intervention arm. The sensitivity analyses revealed robust results. Conclusion: VDS seems not to affect blood lipids in adults with MetS.

## 1. Introduction

Metabolic syndrome (MetS) is a conglomeration of cardiometabolic disorders that collectively increases a person’s risk for developing type 2 diabetes mellitus (T2DM) and cardiovascular disease (CVD) [1,2]. Over the last two decades, the number of people diagnosed with MetS has increased considerably, encompassing 20% to 25% of the adult population and presenting an enormous public health issue [3,4].

The precise definition of MetS varies slightly between guidelines issued by expert groups including the World Health Organization (WHO); the National Cholesterol Education Program Third Adult Treatment Panel (NCEP ATP III); the International Diabetes Federation (IDF); and the American Heart Association/National Heart, Lung, and Blood Institute [5]. Yet, the core components of this syndrome consist of glucose intolerance, hypertension, dyslipidemia—specifically, reduced high-density lipoprotein cholesterol (HDL-C), elevated triglycerides (TG), and central obesity [6].

Individuals with MetS are at an increased risk for CVD, with atherogenic dyslipidemia (low HDL-C and hypertriglyceridemia) being a major underlying cause for its development [7]. Atherogenic dyslipidemia emerges as the greatest competitor of low-density lipoprotein cholesterol (LDL-C) among lipid risk factors for CVD [8,9,10]. Achieving a better understanding of this atherogenic dyslipidemia and factors associated with it may provide clues and further insight into possible interventions that may reduce the risk of CVD in this patient population [5,10].

Vitamin D supplementation (VDS) is among those interventions suggested to alleviate atherogenic dyslipidemia in patients with MetS [11]. Vitamin D is a fat-soluble vitamin that has an integral role in skeletal and immune system disorders [12], along with numerous metabolic functions, including glucose homeostasis, insulin regulation of body weight, and a potent modifier of cardiovascular risk [13]. Vitamin D deficiency, or low levels of 25-hydroxyvitamin D, is associated with a higher risk of MetS. Additionally, suboptimal levels of the vitamin may increase the severity of the syndrome [14,15]. Concentrations of 25-hydroxyvitamin D are lower in patients with MetS compared with those without it [16], and the prevalence of MetS is reduced by half if individuals have high 25-hydroxyvitamin D concentrations [17]. Specifically, vitamin D might modulate the atherogenic components of MetS. A significant inverse relationship has been observed between higher levels of serum 25-hydroxyvitamin D and hypertriglyceridemia, in addition to a positive association with HDL-C [18,19,20]. Nevertheless, some studies report a controversial association between low levels of 25-hydroxyvitamin D and MetS and its individual components [21,22]. Numerous randomized controlled trials (RCTs) have investigated the effect of VDS on dyslipidemia among patients with MetS and found conflicting results [16,23,24]. Therefore, the aim of this systematic review and meta-analysis is to summarize the available evidence of RCTs to establish the impact of VDS on dyslipidemia among adult patients with MetS.

## 2. Materials and Methods

### 2.1. Review Design

The review was conducted according to the Preferred Reporting Items for Systematic Reviews and Meta-Analyses (PRISMA) statement [25] and following a predefined protocol that was registered at the OSF registries (DOI: 10.17605/OSF.IO/XBJM8). Ethical approval was not required for the current study.

### 2.2. Criteria for Study Inclusion

This systematic review included randomized controlled trials (RCTs) conducted on adults with the metabolic syndrome, including an intervention group that received supplementation with vitamin D and a control group, where dyslipidemia was reported as an outcome.

RCTs supplementing vitamin D3 or D2 in any form to the intervention group, and a placebo or a lower dose of vitamin D provided to the control group; investigating at least one of the dyslipidemia components of the metabolic syndrome (Total Cholesterol (TC), LDL-C, HDL-C, or TG) measured in the fasting state; including adult participants, as defined by the investigators—e.g., aged > 18 years at baseline, suffering from the metabolic syndrome (irrespective of the definition adopted)—were included. Only RCTs with a minimum duration of 4 weeks were included to ensure that the intervention had sufficient time to produce an effect. Additionally, RCTs involving a co-intervention were included if both arms of the study received the same co-intervention.

Studies were excluded if they were conducted on healthy participants, or participants with chronic or acute conditions other than the metabolic syndrome, or participants receiving medication known to influence vitamin D metabolism.

### 2.3. Search Strategy

The search strategy considered two key concepts: (1) vitamin D and (2) metabolic syndrome. For each concept, Medical Subject Headings (MeSH) and keywords were mapped. Search terms included but were not limited to vitamin D, cholecalciferol, ergocalciferol, or calcidol, combined with metabolic syndrome. The following databases were searched: PUBMED, MEDLINE, CINAHL, EMBASE, the Cochrane Library, ClinicalTrials.gov, and the International Clinical Trials Registry Platform (ICTRP) [26,27]. No language restrictions were applied to the search; however, the timeline was limited to studies published after the year 1998—when the first definition of the metabolic syndrome was issued by the World Health Organization [3]—until 31 July 2020. The electronic search strategy was validated by a medical information specialist and is described in the Supplement. Bibliographies of included RCTs and relevant reviews were also hand-searched for eligible studies.

### 2.4. Study Selection

Two pairs of authors screened titles and/or abstracts retrieved by the search and identified studies that potentially meet the inclusion criteria outlined above. The two pairs then reviewed the full texts of potentially eligible studies independently and in duplicate, and assessed them for eligibility. To ensure the validity of the study selection process, a calibration exercise was first conducted. Disagreements were solved through consensus or with the help of a third reviewer.

### 2.5. Data Extraction

Two pairs of authors extracted data from eligible studies independently and in duplicate using a data extraction form. A calibration exercise was first conducted to ensure the validity of the data extraction process. For all eligible records, the authors recorded characteristics of the study, details of the population, interventions (type, form, and the dose of vitamin D in experimental groups, comparator, and duration), outcomes assessed, as well the main findings. Serum 25OHD was converted to nmol/L, if it was reported as ng/mL by multiplying by a factor of 2.496. Serum TC, LDL-C, HDL-C, and TG were converted to mmol/, if they were reported as mg/dL, using the respective multiplication factors: 0.0259 for TC, LDL-C, and HDL-C, and 0.0113 for TG.

### 2.6. Quality Assessment

Two pairs of authors assessed independently and in duplicate the risk of bias of included RCTs following the Cochrane criteria (sequence generation, allocation concealment, blinding of participants and outcome assessors, incomplete outcome data, and selective outcome reporting) [28]. Each potential source of bias was graded as low, high, or unclear risk. Disagreements were solved through consensus or with the help of a third reviewer.

The overall quality of the evidence generated by the meta-analysis was assessed according to the Grading of Recommendations Assessment, Development and Evaluation (GRADE) methodology (high risk of bias, imprecision, indirectness, heterogeneity, and publication bias). The evidence was presented using GRADE Evidence Profiles developed in the GRADEpro software [29].

### 2.7. Data Synthesis

When a meta-analysis was not possible, a narrative review of the findings was performed. Meta-analyses were conducted when participants, treatments, and the outcomes were similar enough to allow pooling. Standard meta-analyses comparing VDS with no supplementation in patients with metabolic syndrome were performed using RevMan version 5.3 (The Cochrane Collaboration, The Nordic Cochrane Centre). A fixed-effects model was used when analyzing two studies and a random-effects model when analyzing more than two studies. The results were reported on as a weighted mean difference with 95% confidence intervals. The I^2^ statistic was used to assess heterogeneity among different studies. The I^2^ metric ranges from 0 to 100%, with higher values indicating greater heterogeneity. In cases of moderate to substantial heterogeneity, with I^2^ values greater than 50%, the potential causes were explored and reported on, and relevant sensitivity analyses were conducted.

## 3. Results

### 3.1. Search Results

Details of the search process are presented in Figure 1. Seven studies were included in the systematic review. Out of the seven included studies, four yielded data that could be combined in the meta-analysis.

### 3.2. Characteristics of Included Studies

Characteristics of included studies are given in Table 1. The studies by Makariou [30,31,32] were conducted on the same sample, but reported on different outcomes in three different manuscripts. Three of the studies were conducted in Greece [30,31,32], two in Iran [33,34], one in Thailand [33] and one in China [34]. The number of trial participants varied from 50 to 123, and a mean age ranging between 40 and 65 years. All of the studies were conducted on participants suffering from the MetS, diagnosed either by the NCEP-ATP III [30,31,32,33,34], the IDF [35], or the joint interim statement between several major organizations [36]. The follow-up period varied from 8 weeks [33] to 1 year [34].

In three studies, the intervention consisted of vitamin D3 supplementation with dietary intervention [30,31,32], one study used vitamin D (without specifying its type) supplementation with physical activity [35], two studies supplemented only with vitamin D3 [34,36], and one study supplemented with vitamin D2 [33]. The average daily dose of VDS ranged from 700 IU [34] to 7142.85 IU [36], whereby four studies were supplemented with 2000 IU per day [30,31,32,35]. Four RCTs were placebo-controlled [33,34,35,36], and in the other three RCTs [30,31,32], the comparator was dietary intervention according to the NCEP-ATP III guidelines. Only Yin et al. [34] included a co-intervention in the form of calcium supplementation.

As for study outcomes detailed in Table 2, TC, TG, HDL-C, and LDL-C were analyzed in four studies [30,33,35,36]. In addition, Salekzamani et al. [36] assessed TG/HDL-C and LDL-C/HDL-C. Yin et al. [34] analyzed TG, HDL-C, and LDL-C only. Novel lipid-related biomarkers were also assessed in Makariou et al. [30]—i.e., apo A1 and Apo B; Makariou et al. [32]—i.e., oxidized-LDL, oxidized-LDL/LDL, and oxidized LDL/ApoB; Makariou et al. [31]—i.e., sLDL-C and mean LDL size.

### 3.3. Assessment of Risk of Bias

The assessment of the risk of bias of included studies is presented in Figure 2. The quality of the RCTs design and reporting was low in general and varied across studies. Random allocation of participants was reported in the three studies by Makariou et al. [30,31,32] and in the study by Salekzamani et al. [36], and was unclear in the other three studies [33,34,35]. Only Salekzamani et al. [36] gave sufficient detail to ascertain adequate allocation concealment, while this was unclear in the other studies [30,31,32,33,34,35]. Blinding of participants was impossible in the studies by Makariou et al. [30,31,32], reflecting a high risk of bias, and was guaranteed only in the study by Salekzamani et al. [36] and Wongwiwatthananukit et al. [33]. All trials had adequate blinding of outcome assessment, complete outcome data, and low selective reporting bias.

### 3.4. Results of Included Studies

Table 2 describes the findings from the included studies. All the included studies [30,31,32,33,34,35,36] reported a significant increase in vitamin D status in the intervention groups at endline. Regarding end-point values of lipid parameters, Makariou et al. [30], Wongwiwatthananukit et al. [33], and Yin et al. [34] found no significant differences in TC, TG, LDL-C, and HDL-C between the compared groups. In Farag et al. [35], TG at baseline was significantly higher in the vitamin D group compared with the other groups, and HDL-C was significantly higher in the vitamin D + physical activity group compared with the other groups, which hindered the direct comparison between the three groups at endline. The authors reported that endline TC was significantly lower in the vitamin D group compared with the other groups, LDL-C was significantly lower in the vitamin D group compared with the placebo group, and HDL-C was significantly higher in the vitamin D + physical activity group compared with the other groups. Regarding within-group changes, there was a greater significant decrease in TC and LDL-C in the vitamin D + physical activity group compared with the placebo group; and no other differences in changes in TG and HDL-C between baseline and endline were noted in the three groups [35]. Additionally, in Salekzamani et al. [36], at baseline, TG and TG/HDL-C were significantly higher in the intervention group than the control group. At endline, the authors reported a greater decrease in TG and TG/HDL-C in the vitamin D group compared with the C group, but did not find significant changes in other parameters, namely TC, HDL-C, LDL-C, and LDL-C/HDL-C [36]. Similarly, no significant changes in novel lipid-related biomarkers were noted with VDS in the two studies by Makariou et al. [31,32].

### 3.5. Results of the Meta-Analyses

Two of the studies by Makariou et al. [31,32] and that by Yin et al. [34] were not included in the meta-analysis; as Makariou et al. [31,32] solely reported on novel lipid-related biomarkers, namely oxidized LDL-C and small-density LDL-C (sdLDL-C), and the study by Yin et al. [34] was conducted over the period of one year—a duration that is much longer than the other studies. Moreover, in the study by Farag et al. [35], the intervention arm entailing vitamin D + physical activity was excluded from the meta-analysis since the control arm consisted of administration of placebo only, without physical activity. In contrast, the study by Makariou et al. [30] was included in the analysis since both arms entailed a dietary intervention, allowing it to be canceled out.

Based on the administered daily dose equivalent of vitamin D, two sets of meta-analyses were conducted. The first one included the studies by Makariou et al. [30], Farag et al. [35], and the I^2^ arm of the study by Wongwiwatthananukit et al. [33]. The analysis consisted of comparing a low dose of VDS versus no supplementation, namely placebo or dietary intervention. The other analysis included the study by Salekzamani et al. [36] and the I1 arm of the study by Wongwiwatthananukit et al. [33] and consisted of comparing a high dose of VDS versus placebo.

Forest plots for the mean difference in LDL-C, HDL-C, TC, and TG for the two sets meta-analyses based on the daily dose equivalent of vitamin D in the intervention arms are presented in Figure 3 and Figure 4, respectively. The first set of meta-analyses revealed no statistically significant difference in LDL-C, HDL-C, and TC between patients receiving low-dose VDS compared with those not receiving it. Furthermore, the meta-analysis revealed a statistically significant increase in TG in the group receiving VDS compared with placebo (mean difference, 30.67 (95% CI, 4.89, 56.45) mg/dL; *p* = 0.02). Yet, the heterogeneity of this analysis was substantially high (I^2^ = 86%) (Figure 3). The final quality of evidence of all the meta-analyses was very low (Appendix A). Appendix A presents the results of the sensitivity analyses, which were based on the exclusion of Farag et al. [35], as a source of heterogeneity. The study had inadequate randomization that is reflected in the incomparable baseline TG of the randomized arms. Excluding this study dropped the heterogeneity to none, yet, the sensitivity analyses did not affect the results.

Similarly, the meta-analyses revealed no statistically significant difference in LDL-C, HDL-C, and TC between patients receiving high-dose VDS compared with those receiving a placebo. Additionally, the meta-analysis revealed a statistically significant increase in TG in the group receiving VDS compared with placebo (mean difference, 27.33 (95% CI, 2.06, 52.59) mg/dL; *p* = 0.03) (Figure 4). The heterogeneity of this analysis was also high (I^2^ = 51%). Similar to the first set of meta-analyses, one of the included studies, namely that by Salekzamani et al. [36], had unequal baseline TG levels between the randomized arms. Excluding this study and conducting a sensitivity analysis was impossible, as this set of meta-analyses included only two studies. The final quality of evidence of these meta-analyses was low (Appendix A).

## 4. Discussion

Vitamin D deficiency is a worldwide public health problem that affects all age groups [37]. It is widespread even in sunny countries [38] and in those that have implemented a rigorous VDS strategy for years [1,2]. The high prevalence of vitamin D deficiency is associated with various factors, including genetics, skin pigmentation, latitude, air pollution, obesity, in addition to behavioral lifestyle factors, such as sun avoidance, reduced outdoor activities, and use of sunscreen [39].

In parallel, MetS has recently surfaced as a major public health problem and a leading risk factor for the progression of T2DM and CVD [8,9,10]. Specifically, atherogenic dyslipidemia in MetS emerged as a key factor for CVD and a target for future interventions aiming at reducing poor patient outcomes [7,10]. Patients with MetS were reported to have decreased 25-hydroxyvitamin D levels [40]. Accordingly, correcting vitamin D deficiency through VDS was suggested to alleviate MetS, specifically the atherogenic dyslipidemia component of this syndrome [11]. This topic is gaining attention in the research world and is of clinical relevance [41].

To date, the literature presents conflicting results on the effects of VDS on the dyslipidemia component of MetS. Specifically, observational data indicate an inverse association between hypovitaminosis D and dyslipidemia in patients with MetS [16]. However, our findings indicate that correcting suboptimal vitamin D levels through supplementation was not effective in improving dyslipidemia. VDS, whether as D2 or D3, in a high or low dose, for a short or long duration, although significantly increased vitamin D blood levels, did not significantly affect TC, LDL-C, and HDL-C levels, nor the levels of novel lipid-related biomarkers. Furthermore, VDS significantly increased TG levels compared with placebo, although the baseline TG levels of compared arms in two of the included studies [35,36] were not comparable, which limits this finding. Our results are similar to those of other reviews reporting no meaningful changes in blood lipid values secondary to VDS in healthy, obese, or diabetic subjects [42,43].

The direct or indirect mechanisms through which vitamin D influences the lipid profile remain unclear [44]. Since the observational and interventional studies have conflicting evidence, it has been suggested that the association between vitamin D and metabolic disorders may be confounded by obesity rather than being a causal relationship. Obesity reduces the detectable serum levels of 25-hydroxyvitamin D through the sequestration of vitamin D in body fat tissue or decreased skin synthesis of vitamin D due to the limited outdoor activity and sun exposure [40,45]. Moreover, chronic inflammatory processes, which usually present in obese patients, might decrease 25-hydroxyvitamin D levels [46] and simultaneously affect various metabolic parameters. Accordingly, the relationship between vitamin D deficiency and poor metabolic profile may be explained by the fact that both of these factors are prone to cluster in obese subjects. It is thus possible that high vitamin D levels are not the cause of good health, rather its outcome, since healthy people generally stay outdoors longer and have better eating habits [42].

Furthermore, the dose, frequency, and duration of supplementation with vitamin D might also explain the discouraging results of interventional studies. For instance, supplementation for a period of three months may not be long enough to have a significant effect. The concentration of serum vitamin D would need to be in the range 100–150 nmol/L for cardiovascular disease protection [47], whereas the mean endline vitamin D levels in the intervention groups of the studies included in this review fell well below this level. Furthermore, VDS should be administered on a daily basis to ensure stable circulating concentrations for optimal functioning of the endocrine system [48]. Therefore, short treatment durations and bolus doses of some of the included RCTs could explain the null effects. Finally, it is also possible that vitamin D could provide benefits for cardiometabolic health through improvement in markers other than the lipid profile, such as in endothelial function [49], or through its effect on improving serum calcium profile early in the disease course. The latter observation is suggested by RCTs showing improvements in lipid profile in non-lean healthy subjects with low dietary calcium intake following vitamin D and calcium supplementation [50].

It is worthy to note that, to date, there is no consensus on the most suitable approach to correct vitamin D deficiency, and we lack information on the form, dose, frequency, and duration of vitamin D intervention that would be required to improve the metabolic components of MetS.

Multiple determinants may affect vitamin D status including genetic variation which could have a clinically important impact on response to VDS treatment among different individuals with identical doses [51,52]. For a better understanding of the regulation of vitamin D metabolism and its relation to dyslipidemia, variants of several genes including VDR which encodes the vitamin D receptor, DHCR7 which encodes the enzyme 7-dehydrocholesterol reductase, CYP2R1 which encodes the hepatic enzyme 25- hydroxylase, CYP24A1 which encodes 24-hydroxylase, and GC which encodes DBP the transporting protein for vitamin D DBP should all be considered [53]. Moreover, it is possible that single nucleotide polymorphisms (SNPs) in genotypes could modify the optimal vitamin D status required to reduce MetS disease outcomes [54]. Other confounding factors such as seasonal variation (vitamin D levels rise in summer and drop in winter) and geographic latitude have an important impact on vitamin D status and its correlation with health risk assessment [55]. Since the reviewed RCTs have not examined the genetic predisposition, nor the seasonal effect, it could be misleading to firmly conclude that VDS had no impact on dyslipidemia among MetS patients and hence further investigations are still warranted.

## 5. Strengths and Limitations

To our knowledge, this is the first review to systematically assess the effect of VDS and its effect on dyslipidemia, specifically in adults with MetS. The main strength of our review is that we included only RCTs, which generally had a low risk of bias. Another strength is that we conducted this review according to a predefined protocol, following standard methods for reporting systematic reviews (Moher, 2010), and using a comprehensive and sensitive search strategy with multiple databases and gray literature. We also employed several sensitivity analyses to assess the robustness of our results, whereby in cases of moderate to substantial heterogeneity, we explored and reported on the potential causes. However, our findings are limited by the small number of identified studies, their small sample sizes, and short duration. Furthermore, three of the included studies [32,35,36] started out with significantly higher baseline lipid levels in the intervention group, which limits the results generated by this review and pertaining meta-analyses.

## 6. Conclusions

Physiological mechanisms throughout epidemiological data suggest a link between vitamin D deficiency and MetS. Yet, we report inconsistent results on the relationship between vitamin D status and dyslipidemia in adults with MetS, mainly pointing towards a lack of effect, despite improvement in vitamin D status. Our results should be interpreted with caution given the limited number of included RCTs, the small sample size, and limited intervention period. It is plausible that potentially the associations between vitamin D and cardiometabolic health are not causal; this was also suggested regarding the link between vitamin D and a wide range of acute and other chronic health disorders [56]. Despite the fact that the positive outcome of VDS for improving dyslipidemia among patients with MetS was weak, this does not eliminate the beneficial effect of vitamin D in this subpopulation of patients as an anti-inflammatory hormone which mediates muscle strength and homeostasis [57]. The use of vitamin D status for clinical implications has been well established for many diseases including CVDs [58,59]. Several mendelian randomization studies have supported the protective role of VDS against some diseases such as MS [60]. Hence, further studies are needed before making any solid conclusions about the vitamin D status for clinical implications for dyslipidemia in the context of MetS. Till then, it remains crucial to achieve vitamin D sufficiency in patients with MetS.

## Figures and Tables

**Figure 1 nutrients-12-03352-f001:**
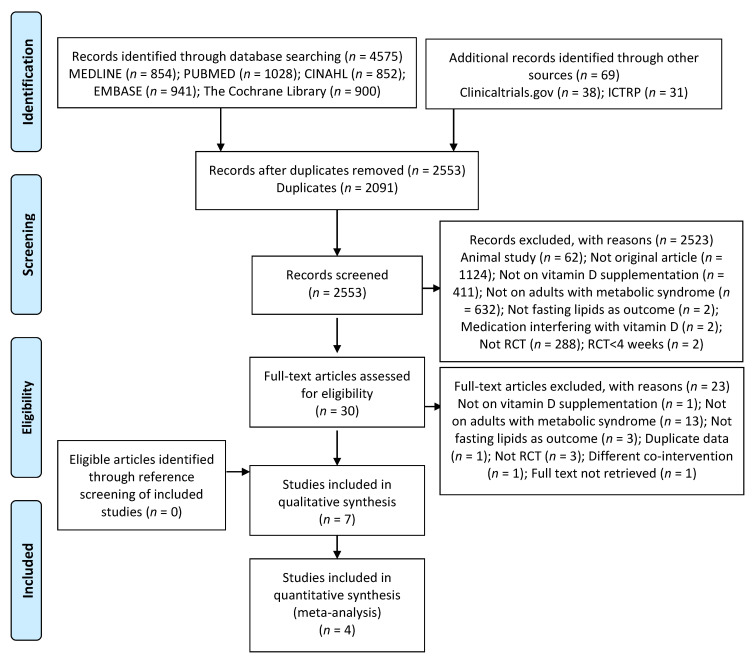
Preferred Reporting Items for Systematic Reviews and Meta-Analyses (PRISMA) diagram of study selection. ICTRP: International Clinical Trials Registry Platform; RCT: Randomized Controlled Trial.

**Figure 2 nutrients-12-03352-f002:**
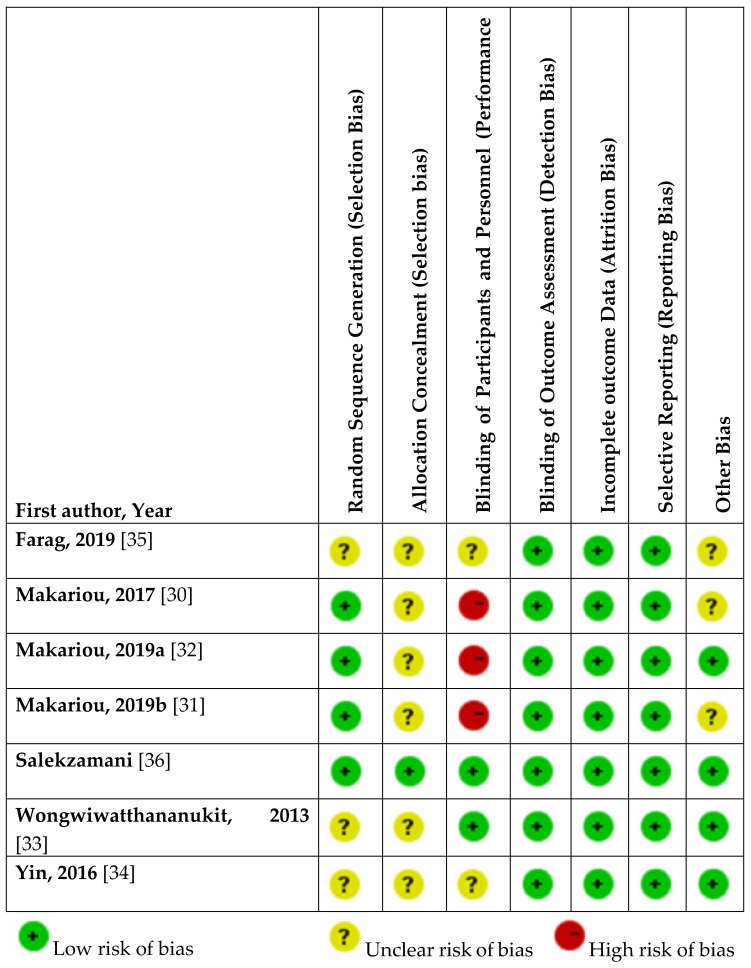
Risk of bias of included studies from consensus between a pair of raters.

**Figure 3 nutrients-12-03352-f003:**
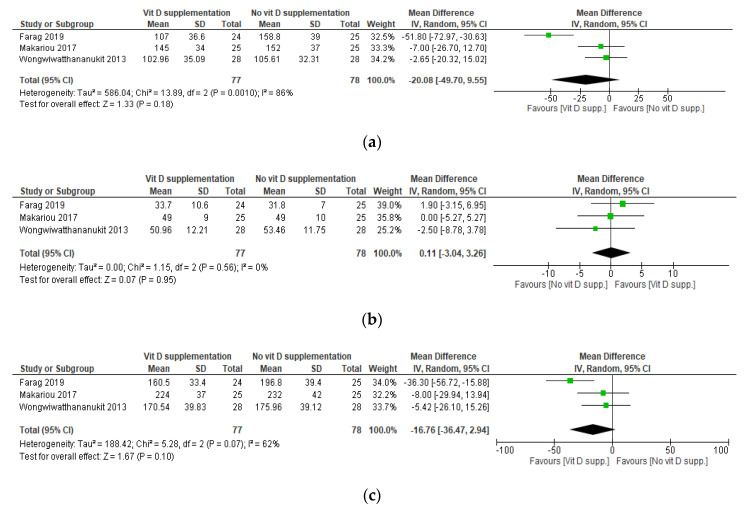
Meta-analysis of effects of low-dose VDS on LDL-C, HDL-C, TC, and TG. Mean differences for each study are represented by squares, and 95% Confidence Intervals are represented by the lines through the squares. The pooled mean differences are represented by diamonds. Between-study heterogeneity was assessed with the use of the I^2^ statistic. VDS: Vitamin D Supplementation; LDL-C: Low-density Lipoprotein Cholesterol; HDL-C: High-density Lipoprotein Cholesterol; TC: Total Cholesterol; TG: Triglycerides. (**a**) Forest plot of mean differences in LDL-C (in mg/dL) between subjects receiving low-dose VDS compared with those not receiving VDS. (**b**) Forest plot of mean differences in HDL-C (in mg/dL) between subjects receiving low-dose VDS compared with those not receiving VDS. (**c**) Forest plot of mean differences in TC (in mg/dL) between subjects receiving low-dose VDS compared with those not receiving VDS. (**d**) Forest plot of mean differences in TG (in mg/dL) between subjects receiving low-dose VDS compared with those not receiving VDS. * The study by Makariou et al. [30] was excluded from the primary analysis since the data are reported as median and range. The median cannot be assumed the same as the mean, and the standard deviations cannot be extrapolated from the range since the sample size is small. In addition, the study explicitly reports on the use of median and range when the distribution is skewed. * In the study by Farag et al. [35], TG at baseline was significantly higher in the intervention group compared with the control group.

**Figure 4 nutrients-12-03352-f004:**
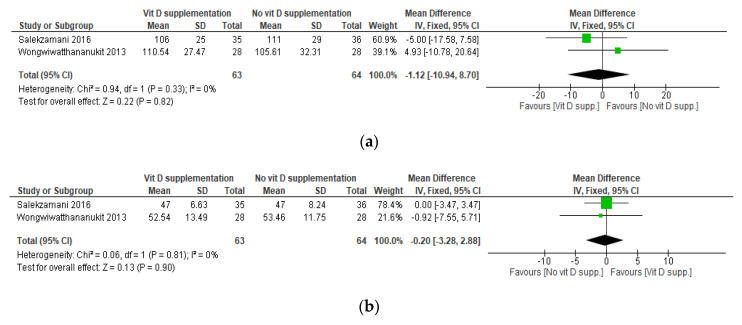
Meta-analysis of effects of high-dose VDS on LDL-C, HDL-C, TC, and TG. Mean differences for each study are represented by squares, and 95% Confidence Intervals are represented by the lines through the squares. The pooled mean differences are represented by diamonds. Between-study heterogeneity was assessed with the use of the I^2^ statistic. VDS: Vitamin D Supplementation; LDL-C: Low-density Lipoprotein Cholesterol; HDL-C: High-density Lipoprotein Cholesterol; TC: Total Cholesterol; TG: Triglycerides. (**a**) Forest plot of mean differences in LDL-C (in mg/dL) between subjects receiving a high dose of VDS compared with those not receiving VDS. (**b**) Forest plot of mean differences in HDL-C (in mg/dL) between subjects receiving a high dose of VDS compared with those not receiving VDS. (**c**) Forest plot of mean differences in TC (in mg/dL) between subjects receiving a high dose of VDS compared with those not receiving VDS. (**d**) Forest plot of mean differences in TG (in mg/dL) between subjects receiving a high dose of VDS compared with those not receiving VDS. * The mean differences of the two studies are very different and the heterogeneity is 51%. This might be due to the study by Salekzamani et al. [36] since TG at baseline was significantly higher in the intervention group compared with the control group.

**Table 1 nutrients-12-03352-t001:** Characteristics of included studies.

First Author, Year	Study Design	Geographic Setting/Data Collection Time Period	Study Population	Definition of Metabolic Syndrome	Intervention	Dose, Frequency, Duration	Daily Dose Equivalent	Control	Co-Intervention	Compliance	Drop-Out
Farag, 2019 [35]	Parallel randomized placebo-controlled trial	Halabja (Kurdistan Region of Iraq)/March to May	I1: *n* = 24; I2: *n* = 21; C: *n* = 25 Ethnicity: NR Mean age (SD): I1: 40.54 (5.94); I2: 40.42 (5.89); C: 42.6 (5.62) %Male: I1: 33.3%; I2: 33.3%; C: 52.0%	IDF criteria	I1. Vitamin D without PA I2. Vitamin D + PA	I1: 2000 IU, Daily, 12 weeks I2: 2000 IU, Daily, 12 weeks + 30 min of endurance PA, Daily	2000 IU	Placebo without endurance PA	None	NR	I1: 20% I2: 30% C: 16.66%
Makariou, 2017 [30]	Prospective, randomized, open-label, blinded end-point trial	Greece/March to September	I: *n* = 25; C: *n* = 25 Ethnicity: NR Mean age (SD): I: 52 (9); C: 51 (12) %Male: I: 60%; C: 44%	NCEP-ATP III criteria	Vitamin D3 + dietary intervention according to the NCEP-ATP III guidelines	2000 IU, Daily, 12 weeks	2000 IU	Dietary intervention according to NCEP-ATP III guidelines	None	Compliance with Vitamin D: NR Poor compliance with dietary instructions in both groups	0%
Makariou, 2019a [32]	Prospective, randomized, open-label, blinded end-point trial	Greece/March to September	I: *n* = 25; C: *n* = 25 Ethnicity: NR Mean age (SD): I: 53 (7); C: 52 (15) %Males: I: 60%; C: 40%	NCEP-ATP III criteria	Vitamin D3 + dietary intervention according to NCEP-ATP III guidelines	2000 IU, Daily, 12 weeks	2000 IU	Dietary intervention according to NCEP-ATP III guidelines	None	I: 100%; poor compliance with dietary instructions C: Poor compliance with dietary instructions	0%
Makariou, 2019b [31]	Prospective, randomized, open-label, blinded end-point trial	Greece/March to September	I: *n* = 25; C: *n* = 25 Ethnicity: NR Mean age (SD): I: 53 (7); C: 52 (15) %Males: I: 60%; C: 40%	NCEP-ATP III criteria	Vitamin D3 + dietary intervention according to NCEP-ATP III guidelines	2000 IU, Daily, 12 weeks	2000 IU	Dietary intervention according to NCEP-ATP III guidelines	None	Compliance with Vitamin D: NR Poor compliancewith dietary instructions in both groups	0%
Salekzamani, 2016 [36]	Randomized placebo-controlled, double-blind parallel trial	Tabriz, Iran/October to June	I: *n* = 35; C: *n* = 36 Ethnicity: NR Mean age (SD): 40.49 (5.04) %Males: 49% (data per group: NR)	Criteria of the joint interim statement of the IDF task force on epidemiology and prevention; NHLBI; AHA; World Heart Federation; International Atherosclerosis Society; and International Association for the Study of Obesity	Vitamin D3	50,000 IU, Weekly, 16 weeks	7142.85 IU	Placebo	None	97% in both groups	I: 12.5% C: 10%
Wongwiwatthananukit, 2013 [33]	Prospective randomized, double-blind, double-dummy, parallel trial	Bangkok, Thailand/January to September	I1: *n* = 28; I2: *n* = 28; C: *n* = 28 Ethnicity: NR Mean age (SD): I1: 62.29 (10.63); I2: 63.61 (13.25); C: 65.07 (11.31) %Male: I1: 53.3%; I2: 50%; C: 50%	NCEP-ATP III criteria	I1: vitamin D2 I2: vitamin D2	I1: 40,000 IU, Weekly, 8 weeks I2: 20,000 IU, Weekly, 8 weeks + 1 placebo capsule, Weekly, 8 weeks	I1: 5714.28 IU I2: 2857.14 IU	Placebo	None	100% in the 3 groups	I1: 6.66% I2: 6.66% C: 6.66%
Yin, 2016 [34]	Randomized placebo-controlled intervention trial	Jinan, North China/November to February	I: *n* = 61; C: *n* = 62 with vitamin D deficiency (25(OH)D < 50 nmol/L) Ethnicity: Northern Chinese Mean age (SD): 49.5 (8.72) %Male: 54% (data per group: NR)	Updated NCEP-ATP III criteria for Asian Americans	Vitamin D3	700 IU, Daily, 1 year	700 IU	Placebo	600 mg elemental Calcium (CalciumCitrate), Daily	95% in both groups	I: 3.17% C: 1.58%

I: Intervention; C: Control; NR: Not Reported; SD: Standard Deviation; 25(OH)D: 25-Hydroxyvitamin D; NCEP-ATP: National Cholesterol Education Program Adult Treatment Panel III; IDF: International Diabetes Federation; NHLBI: National Heart, Lung, and Blood Institute; AHA: American Heart Association; PA: Physical Activity; IU: International Unit.

**Table 2 nutrients-12-03352-t002:** Outcomes and results of included studies.

First Author, Year	Assessment Method: Vitamin D	Assessment Method: Dyslipidemia Outcomes	Baseline 25OHD Level (nmol/L) ng/mL * 2.496 = nmol/L	Endline 25OHD Level (nmol/L) ng/mL * 2.496 = nmol/L	Baseline Dyslipidemia Outcomes HDL-C, LDL-C: mmol/L * 38.67 = mg/dL TG: mmol/L * 88.57 = mg/dL	Endline Dyslipidemia Outcomes HDL-C, LDL-C: mmol/L * 38.67 = mg/dL TG: mmol/L * 88.57 = mg/dL	Conclusion
Farag, 2019 [35]	25(OH)D: measured by immunoassay	TC, TG: measured using enzymatic colorimetric tests HDL-C: measured after precipitation of the apolipoprotein B containing lipoproteins with phosphotungistic acid LDL-C: calculated from serum TC, TG and HDL-C based on relevant formula	Mean (SD) I1: 26.70 (6.98) I2: 25.95 (7.98) C: 30.20 (9.73)	Mean (SD) I1: 57.90 (12.23) I2: 72.38 (13.72) C: 31.44 (9.98)	TC mean (SD) (mg/dL) I1: 173.5 (60.8) I2: 194.7 (32.2) C: 185.9 (39) HDL-C mean (SD) (mg/dL) I1: 34.9 (17.3) I2: 40.9 (14.4) C: 30.04 (8.5) LDL-C mean (SD) (mg/dL) I1: 120.7 (64.4) I2: 149.6 (35.8) C: 150.4 (39.8) TG mean (SD) (mg/dL) I1: 229.3 (113.8) I2: 184.5 (98.5) C: 174.4 (43)	TC mean(SD) (mg/dL) I1: 160.5(33.4) I2: 181.7(31.3) C: 196.8(39.4) HDL-C mean (SD) (mg/dL) I1: 33.7 (10.6) I2: 39 (10) C: 31.8 (7) LDL-C mean (SD) (mg/dL) I1: 107(36.6) I2: 138.3(31.4) C: 158.8(39) TG mean (SD) (mg/dL) I1: 233.8 (97) I2: 178.1 (80.8) C: 158.6 (35.4)	At baseline, TG was significantly higher in the I1 compared with the other groups; and HDL-C was significantly higher in the I2 group compared with the other groups There were NS differences in other study parameters between groups At endline, 25(OH)D was significantly higher in the I1 and I2 group compared with the C group TC was significantly lower in the I1 group compared with the other groups LDL-C was significantly lower in the I1 group compared with the C group HDL-C was significantly higher in I2 compared with the other groups Greater significant decrease in TC and LDL-C in I2 compared with the C group There were NS differences in changes in TG and HDL-C between baseline and endline in the 3 groups
Makariou, 2017 [30]	25(OH)D: measured by enzyme immunoassay	TC, TG, HDL-C: measured enzymatically LDL-C: calculated by the Friedewald equation (when TG < 350 mg/dl) ApoA1, ApoB: measured by immunonephelometry	median (min–max) I: 39.93 (7.48–87.36) C: 24.96 (9.98–117.84)	median (min–max) I: 76.37 (20.96–167.23) C: 32.44 (8.73–92.35)	TC mean (SD) (mg/dL) I: 219 (36) C: 231 (34) HDL-C mean (SD) (mg/dL) I: 48 (10) C: 50 (9) LDL-C mean (SD) (mg/dL) I: 140 (35) C: 147 (26) TG median (min–max) (mg/dL) I: 150 (56–336) C: 146 (84–339) Apo A1 mean (SD) (mg/dL) I: 136 (26) C: 143 (13) Apo B mean (SD) (mg/dL) I: 92 (25) C: 107 (16)	TC mean (SD) (mg/dL) I: 224 (37) C: 2232 (42) HDL-C mean (SD) (mg/dL) I: 49 (9) C: 49 (10) LDL-C mean (SD) (mg/dL) I: 145 (34) C: 152 (37) TG median (min-max) (mg/dL) I: 136 (46–261) C: 131 (73–307) Apo A1, ApoB: NR	At baseline, there were NS differences in study parameters between groups At endline, 25(OH)D was significantly higher in the I group compared with the C group There were NS differences in lipid parameters between groups
Makariou, 2019a [32]	25(OH)D: measured by enzyme immunoassay	Oxidized-LDL: measured by a competitive enzyme-linked immunosorbent assay using a specific murine monoclonar antibody Oxidized-LDL/LDL: NR Oxidized-LDL/ApoB: NR	median (95%CI) I: 40.18 (25.70–61.90) C: 24.71 (13.72–39.18)	median (95%CI) I: 76.37 (64.14–103.58) C: 32.94 (19.96–58.65)	Oxidized LDL-C mean (SD) (95%CI) (U/L) I: 70.3 (15.2) (64.8–87.6) C: 67.2 (16.9) (59.2–79.3) Oxidized LDL-C/LDL-C mean (SD) (95%CI) (U/mg) I: 0.05 (0.01) (0.46–0.65) C: 0.06 (0.008) (0.41–0.58) Oxidized LDL-C/ApoB mean(SD) (95%CI) (U/mg) I: 0.08 (0.04) (0.70–1.08) C: 0.07 (0.008) (0.57–0.74)	Oxidized LDL-C mean(SD) (95%CI) (U/L) I: 75.9(21.2) (67.9–89.9) C: 67.3(19.3) (56.9–92.5) Oxidized LDL-C/LDL-C mean(SD) (95%CI) (U/mg) I: 0.05(0.01) (0.46–0.60) C: 0.06(0.02) (0.42–0.78) Oxidized LDL-C/ApoB mean(SD) (95%CI) (U/mg) I: 0.07(0.01) (0.69–0.82) C: 0.08(0.02) (0.66–0.91)	At baseline, Ox-LDL/ApoB (U/mg) was significantly higher in the C group compared with the I group There were NS differences in other study parameters between groups At endline, 25(OH)D was significantly higher in the I group compared with the C group There were NS differences in lipid parameters between groups
Makariou, 2019b [31]	25(OH)D: measured by enzyme immunoassay	sdLDL-C: analyzed electrophoretically sdLDL proportion, mean LDL size: analyzed using the methods of the European Panel On Low-Density Lipoprotein Subclasses	median (min–max) I: 40.18 (8.23–87.60) C: 24.71 (9.98–98.84)	median (min–max) I: 76.37 (20.96–168.72) C: 32.94 (8.73–91.85)	sdLDL median (min–max) (mg/dL) I: 9.0 (0.0–40) C: 7.0 (0.0–22) sdLDL proportion mean (SD) (%) I: 5.7 (5.2) C: 3.8 (2.8) LDL size mean (SD) (nm) I: 264.8 (6.3) C: 266.5 (3.9)	sdLDL median (min-max) (mg/dL) I: 4.0 (0.0–46) C: 5.0 (2.0–25) sdLDL proportion mean (SD) (%) I: 4.5 (4.4) C: 3.3 (2.3) LDL size mean (SD) (nm) I: 266.6 (5.2) C: 267.0 (3.5)	At baseline, there were NS differences in study parameters between groups At endline, 25(OH)D was significantly higher in the I group compared with the C group There were NS difference in lipid parameters between groups
Salekzamani, 2016 [36]	25(OH)D: measured by chemiluminescent immunoassay	TG, TC, LDL-C, HDL-C: measured enzymatically TG/HDL-C: NR LDL/HDL-C: NR	Mean (SD) I: 16.45 (15.50) C: 23.47 (21.34)	Mean (SD) I: 78.38 (21.71) C: 21.46 (17.74)	TC mean (SD) (mg/dL) I: 212 (42) C: 200 (39.27) HDL-C mean (SD) (mg/dL) I: 45 (8.08) C: 45 (10.08) LDL-C mean (SD) (mg/dL) I: 114 (33) C: 117 (28) TG mean (SD) (mg/dL) I: 269 (97) C: 185 (61) TG/HDL-C mean (SD) I: 6.05 (2.21) C: 4.22 (1.64) LDL-C/HDL-C mean (SD) I: 2.53 (0.69) C: 2.57 (0.54)	TC mean (SD) (mg/dL) I: 203.21 (34.63) C: 197.14 (33.57) HDL-C mean (SD) (mg/dL) I: 47 ± 6.63 C: 47 ± 8.24 LDL-C mean (SD) (mg/dL) I: 106 ± 25 C: 111 ± 29 TG mean(SD) (mg/dL) I: 242 ± 82 C: 196 ± 72 TG/HDL-C mean (SD) I: 5.20 ± 1.67 C: 4.37 ± 1.99 LDL-C/HDL-C mean (SD) I: 2.27 ± 0.53 C: 2.36 ± 0.64	At baseline, TG and TG/HDL-C were significantly higher in the I group compared with the C group There were NS differences in other study parameters between groups At endline, 25(OH)D significantly increased in the I group and was stable in the C group TG and TG/HDL-C had a greater% change in the I compared with the C group There were NS differences in other lipid parameters between groups
Wongwiwatthananukit, 2013 [33]	25(OH)D: measured by chemiluminescent immunoassay	TC, TG, HDL-C, LDL-C: NR	Mean (SD) I1: 35.66 (8.36) I2: 37.63 (7.88) C: 40.43 (7.46)	Mean (SD) I1: 75.95 (17.39) I2: 66.89 (15.89) C: 47.39 (16.74)	TC mean (SD) (mg/dL) I1: 180.36 (34.43) I2: 166.89 (20.95) C: 174.29 (38.90) HDL-C mean(SD) (mg/dL) I1: 53.18(12.46) I2: 52.36(11.86) C: 53.43(12.73) LDL-C mean(SD) (mg/dL) I1: 107.00(27.46) I2: 96.68(19.96) C: 102.50(29.51) TG mean(SD) (mg/dL) I1: 139.32(61.26) I2: 132.29(62.36) C: 129.46(59.75)	TC mean (SD) (mg/dL) I1: 182.04 (31.00) I2: 170.54 (39.83) C: 175.96 (39.12) HDL-C mean (SD) (mg/dL) I1: 52.54 (13.49) I2: 50.96 (12.21) C: 53.46 (11.75) LDL-C mean (SD) (mg/dL) I1: 110.54 (27.47) I2: 102.96 (35.09) C: 105.61 (32.31) TG mean (SD) (mg/dL) I1: 144.82 (64.07) I2: 137.79 (53.48) C: 135.75 (71.40)	At baseline, there were NS differences in study parameters between groups At endline, 25(OH)D was significantly higher in the I1 and I2 groups compared with the C group There were NS differences in lipid parameters between groups
Yin, 2016 [34]	25(OH)D: measured by double antibody radioimmunoassay	TG, HDL-C: measured by enzymatic colorimetric assay LDL-C: calculated using the Friedwald equation	Mean (SD) I: 36.44 (5.44) C: 35.44 (6.36)	Mean (SD) I: 82.61 (10.90) C: 36.44 (6.98)	HDL-C mean (SD) (mg/dL) I: 41.38 (3.09) C: 38.28 (2.70) LDL-C mean (SD) (mg/dL) I: 126.06 (9.66) C: 123.74 (7.73) TG mean (SD) (mg/dL) I: 295.84 (60.23) C: 280.8 (35.43)	HDL-C mean (SD) (mg/dL) I: 42.15 (2.70) C: 40.22 (2.32) LDL-C mean (SD) (mg/dL) I: 122.97 (9.28) C: 121.42 (8.50) TG mean (SD) (mg/dL) I: 250.66 (36.31) C: 255.1 (19.48)	At baseline, there were NS differences in study parameters between groups At endline, 25(OH)D significantly increased in the I group and was stable in the C group There were NS differences in lipid parameters between groups Similar results were obtained in the obesity and non-obesity subgroups

I: Intervention; C: Control; NR: Not Reported; SD: Standard Deviation; 25(OH)D: 25-Hydroxyvitamin D; min: minimum; max: maximum; TC: Total Cholesterol; TG: Triglycerides; HDL-C: High-Density Lipoprotein Cholesterol; LDL-C: Low-Density Lipoprotein Cholesterol; Apo: Apolipoprotein; sdLDL-C: Small Dense Low-Density Lipoprotein Cholesterol; CI: Confidence Interval; NS: non-significant. *: symbol denoting multiplication.

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
