# Peer review of "Effects of Vitamin D Supplementation on Lipid Profile in Adults with the Metabolic Syndrome: A Systematic Review and Meta-Analysis of Randomized Controlled Trials"

_nutrients, 2020, doi:10.3390/nu12113352_

Round 1
Reviewer 1 Report
The study is well designed, elaborately described, and the conclusion is supported by the data. However, the conclusion of the review article state that the results do not support VDS in clinical practice for improving dyslipidemia in MetS patient is too assertive to conclude as many confounding factors has not been taken into consideration. For e.g., what was the time period of the included RCTs (during the year, as weather affects vitD levels and the included countries have different weathers); vitD acts through VDR and the presence of VDR polymorphism may affect the metabolism of vitD; polymorphism f the vitD metabolic enzymes (Cyp24 and Cyp27), liver, and renal status may affect the outcome; what was the absorption status in the patients? was there any disease affecting absorption? The authors have mentioned some of the confounding factors and are suggested to discuss above. Please tone down the conclusion statement to not use vitD in clinics. VitD may not have effects on lipids levels as per the results of this study, it has effects in attenuating inflammation, increasing musculoskeletal strength, and other hormonal effects to maintain homeostasis in case of vitD deficiency or vitD sufficient population with MetS.
Please check few typos e.g., line 323 and 333
Author Response
We thank the reviewer for thoroughly reviewing our manuscript and for his/her valuable comments.
- The study is well designed, elaborately described, and the conclusion is supported by the data. However, the conclusion of the review article state that the results do not support VDS in clinical practice for improving dyslipidemia in MetS patient is too assertive to conclude as many confounding factors has not been taken into consideration. For e.g., what was the time period of the included RCTs (during the year, as weather affects vitD levels and the included countries have different weathers); vitD acts through VDR and the presence of VDR polymorphism may affect the metabolism of vitD; polymorphism f the vitD metabolic enzymes (Cyp24 and Cyp27), liver, and renal status may affect the outcome; what was the absorption status in the patients? was there any disease affecting absorption? The authors have mentioned some of the confounding factors and are suggested to discuss above.
Reply #1: We have modified the conclusion and explained that further studies are needed. We highlighted the confounding factors that might have contributed to the findings including seasonal variation and genetic polymorphism of the vitamin D receptor (VDR) and vitamin D metabolic enzymes. All RCTs with patients suffering from underlying diseases including malabsorption (Crohn’s and Celiac) had been excluded from the SR and hence we did not elaborate on the absorption status of patients. Yet this does not rule out the possibility that patients with abnormal absorption were among the included studies reviewed and could hence have affected the results. Therefore, this factor was also emphasized in the discussion section. We added details on data collection time period in Table 1. In addition, the following had been added to the discussion section on page 28, lines 336-349: “Multiple determinants may affect vitamin D status including genetic variation which could have a clinically important impact on response to VDS treatment among different individuals with identical doses [49,50]. For a better understanding of the regulation of vitamin D metabolism and its relation to dyslipidemia, variants of several genes including VDR which encodes the vitamin D receptor, DHCR7 which encodes the enzyme 7-dehydrocholesterol reductase, CYP2R1 which encodes the hepatic enzyme 25- hydroxylase, CYP24A1 which encodes 24-hydroxylase and GC which encodes DBP the transporting protein for vitamin D DBP should all be considered [51]. Moreover, it is possible that single nucleotide polymorphisms (SNPs) in genotypes could modify the optimal vitamin D status required to reduce MetS disease outcomes [52]. Other confounding factors like seasonal variation (vitamin D levels rise in summer and drop in winter) and geographic latitude have an important impact on vitamin D status and its correlation with health risk assessment [53]. Since the reviewed RCTs have not examined the genetic predisposition, nor the seasonal effect, it could be misleading to firmly conclude that VDS had no impact on dyslipidemia among MetS patients and hence further investigations are still warranted”.
2.Please tone down the conclusion statement to not use vitD in clinics. VitD may not have effects on lipids levels as per the results of this study, it has effects in attenuating inflammation, increasing musculoskeletal strength, and other hormonal effects to maintain homeostasis in case of vitD deficiency or vitD sufficient population with MetS.
Reply #2: Even though our findings did not reveal a significant positive outcome for VDS in improving dyslipidemia among MetS patients, we have acknowledged the well- established protective effect for VDS in reducing inflammation and maintaining muscle strength and homeostasis in this subpopulation. We have modified our conclusion in the abstract. We have also removed the statement that VDS can not be used in clinics and added that further studies are needed to make solid conclusions about the utility of vitamin D in clinical setting on page 29, lines 336-337. The below had been added to the conclusion section on page 29, lines 371-379: “Despite the fact that the positive outcome of VDS for improving dyslipidemia among patients with MetS was weak, this does not eliminate the beneficial effect of vitamin D in this subpopulation of patients as an anti-inflammatory hormone which mediates muscle strength and homeostasis [55]. The use of vitamin D status for clinical implications has been well established for many diseases including CVDs [56,57]. Several mendelian randomization studies have supported the protective role of VDS against some disease like MS [58]. Hence, further studies are needed before making any solid conclusions about the vitamin D status for clinical implications for dyslipidemia in the context of MetS. Till then, it remains crucial to achieve vitamin D sufficiency in patients with MetS”.
3.Please check few typos e.g., line 323 and 333.
Reply #3: We have conducted a thorough proof read and all mistakes had been fixed including the typos on lines 323 and 333.

Reviewer 2 Report
I have read th epaper entitled "Effects of Vitamin D Supplementation on Lipid Profile in Adults with the Metabolic Syndrome: A Systematic Review and Meta‑analysis of Randomized Controlled Trials" by Fatme AlAnouti and coworkers.
The conclusion of the meta analyses is that VDS does not affect blood lipids in adults with MetS.
The paper is well conducted and the criteria of selection of the studied well described and well done.
This reviewer suggests to Author to carefully review the English language.
Author Response
We thank the reviewer for his/her evaluation of the manuscript, as well as for his/her suggestion.
The conclusion of the meta analyses is that VDS does not affect blood lipids in adults with MetS.
The paper is well conducted and the criteria of selection of the studied well described and well done.
This reviewer suggests to Author to carefully review the English language.
Reply: The manuscript was revised by an English language expert. Language errors and proof reading had been addressed, and modifications were done using Track Changes.